# ArabiDoc: A Holistic Arabic-English Evaluation Suite for End-to-End Document Processing

## Abstract

Document intelligence sits at the intersection of computer vision and natural language processing, where the goal is to transform complex real-world documents into structured, machine-readable representations. While progress has been made, current benchmarks for low-resource languages such as Arabic remain limited, typically emphasizing individual components, like text, tables, or charts, without providing a comprehensive evaluation of full-document parsing. To address this gap, we present a new bilingual (Arabic-English) benchmark that brings together diverse document elements within a single evaluation framework for end-to-end document parsing. Our benchmark offers three main contributions. First, it preserves reading order information, which allows models to better capture the natural flow of documents. Second, it supports visual content parsing, encompassing not only text blocks but also tables, charts, and figures, thereby reflecting the full range of document structures. Third, it introduces relaxed evaluation metrics that more fairly assess model performance by tolerating minor deviations in reading order or localized errors in table and chart parsing, ensuring the evaluation reflects practical usability rather than strict exactness. Constructed through a two-step annotation process, layout segmentation followed by object-level labeling, our dataset includes 137 pages that have been carefully segmented and verified by human annotators. By unifying previously separate evaluation tracks, this benchmark establishes the first comprehensive standard for structured document parsing in Arabic and provides a more realistic basis for bilingual evaluation with English. We expect this resource to foster progress in multimodal reasoning, enable stronger baselines, and support the development of vision-language models that generalize robustly across languages and document types.

## 1 Introduction

Extracting structured content from documents, commonly referred to as document parsing, is a central task in modern computer vision and document intelligence. The objective is to transform raw documents into machine-readable representations by capturing both textual information and visual elements such as diagrams and charts. Once extracted, this structured content can directly support downstream applications in context-assisted Large Language Models (LLMs) generation and Multimodal Retrieval-Augmented Generation (RAG) pipelines, thereby enabling tasks such as question answering, document summarization, data analytics, etc.

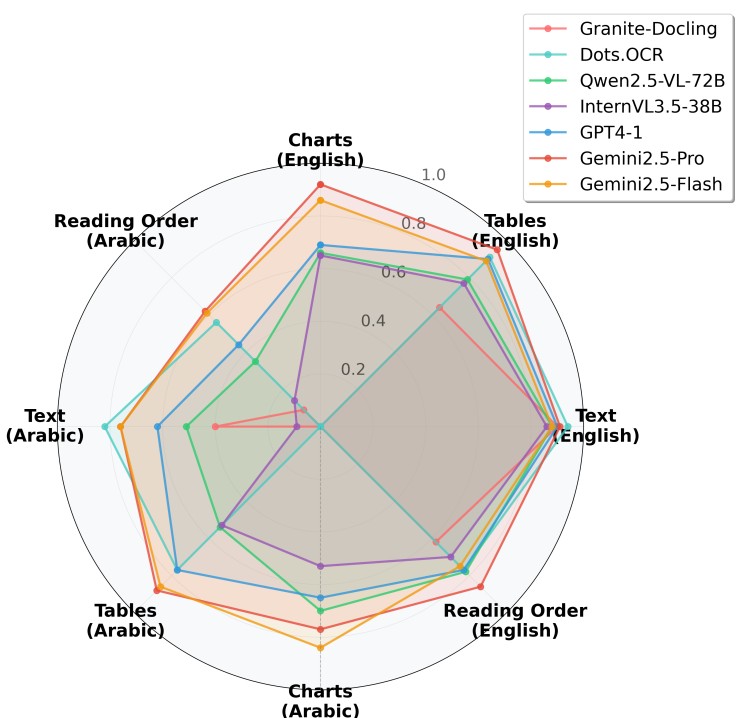

Figure 1: Radar chart comparing English and Arabic full-page parsing results across key components: tables, charts, text, and reading order.

In the research literature, two primary architectural paradigms for document parsing models exist:

- **Pipeline-based approaches** decompose the parsing process into sequential modules, including layout detection (Huang et al., 2022; Zhao et al., 2024), optical character recognition (OCR) (Li et al., 2022; 2021), table/chart recognition (Mustafa et al., 2023; Luo et al., 2021; Li et al., 2025), equation recognition (Li et al., 2020; Wang et al., 2024; Liu et al., 2022), and final assembly of extracted content into the desired output format.

- **End-to-end approaches** leverage VLMs (Blecher et al., 2023; Hu et al., 2024; Bai et al., 2023; Liu et al., 2024; Wei et al., 2024; Bai et al., 2025) trained on large-scale multimodal data to parse complex document pages holistically. These models display very promising results in document parsing capability, even without explicit fine-tuning on a downstream task. VLM models perform the entire content parsing workflow either in a single forward pass or across multiple stages (Feng et al., 2025), but consistently within a unified model.

Both paradigms have demonstrated strong performance in parsing complex and heterogeneous documents. Correspondingly, numerous benchmarks have been developed to evaluate the document parsing capabilities of the above-mentioned models. The limitations of current benchmarks can be articulated on two levels. First, the majority of existing resources predominantly target high-resource languages. While some benchmarks claim multilingual coverage, such as dots.ocr-bench (HiLab, Rednote), these remain inaccessible to the public, thereby reinforcing the need for a dedicated benchmark centered on low-resource languages, such as Arabic. Second, even widely adopted benchmarks like OmniDocBench (Ouyang et al., 2025), despite encompassing diverse document layouts and types, largely omit the evaluation of visual information comprehension within document parsing. This omission is particularly restrictive given that visual elements constitute a substantial portion of critical content in business, finance, and corporate/government reports (e.g., annual reports, executive summaries). Moreover, evaluations of end-to-end parsing models, such as smolDocling and DotsOCR (Nassar et al., 2025; HiLab, Rednote), typically assess chart and table recognition as isolated subtasks rather than as components of holistic full-page annotation. As a result, these approaches fail to capture the interdependencies between textual, structural, and visual modalities that are essential for robust document understanding in real-world applications.

Table 1: A comparison between ArabiDoc and existing Arabic-oriented and multilingual document benchmarks with respect to task coverage and evaluation scope. TR - Table Recognition and CR - Chart Recognition. The End-to-End column represents whether the benchmark supports full-page annotation evaluation that jointly considers text, structural, and visual elements.

| Benchmark | Language | Single-Task Eval | | | End-to-End |
|---|---|---|---|---|---|
| | | OCR | TR | CR | |
| SARD (Nacar et al., 2025) | Arabic | ✓ | × | × | × |
| Arabic-Nougat (Rashad, 2024) | Arabic | ✓ | × | × | × |
| Khatt (Ahmad et al., 2017) | Arabic | ✓ | × | × | × |
| Camel-Bench (Ghaboura et al., 2024) | Arabic | ✓ | × | × | × |
| KITAB-Bench (Heakl et al., 2025) | Arabic | ✓ | ✓ | ✓ | × |
| OmniDocBench (Ouyang et al., 2025) | English & Chinese | ✓ | ✓ | × | ✓ |
| **ArabiDoc** (Ours) | Arabic & English | ✔ | ✔ | ✔ | ✔ |

To address this gap, we introduce **ArabiDoc**, a benchmark designed to holistically evaluate document parsing across English and Arabic languages. Our goal is to establish an evaluation standard for the document parsing task that provides the first comprehensive end-to-end evaluation benchmark incorporating both text and visual modalities.

## 2 RELATED WORKS

The development of robust and accurate document parsing models necessitates a deep understanding of both the visual and textual content presented in real-world documents. Earlier benchmarks, however, exhibited significant limitations in this regard, as they typically targeted only one aspect of the broader and inherently complex task of document parsing. These efforts were largely shaped by the pipelined model design traditionally adopted in this field, where dedicated modules were designed to address specific subtasks, such as layout detection, OCR, equation recognition, and chart interpretation.

With the emergence of end-to-end vision-language models (VLMs), it has now become feasible to unify these subtasks within a single framework. Such models enable comprehensive document parsing that jointly encompasses OCR, layout understanding, table recognition, chart interpretation, and related challenges. This paradigm shift highlights the pressing need for benchmarks capable of evaluating the ability of modern models to extract structured information holistically from complex document pages. In this direction, Ouyang et al. (2025) introduced the OmniDocBench benchmark, which consolidates multiple document parsing tasks into a single evaluation framework and covers documents in English and Chinese languages.

Table 1 provides an overview of benchmark datasets relevant to both individual document parsing subtasks (e.g., OCR, table extraction, chart recognition) and complete end-to-end evaluations. For completeness, we also list benchmarks developed for languages beyond Arabic to provide a broader reference point. As the comparison illustrates, most Arabic-focused benchmarks remain constrained to isolated subtasks (Rashad, 2024; Ghaboura et al., 2024; Ahmad et al., 2017; Heakl et al., 2025; Nacar et al., 2025), with the majority emphasizing OCR and offering little to no coverage of higher-level tasks like layout or visual content recognition. Only Heakl et al. (2025) attempts to extend beyond single-task evaluation by introducing the *PDF-to-Markdown* benchmark. However, closer analysis reveals that this benchmark omits crucial elements such as charts and diagrams, which are essential for capturing the rich semantic content of real-world documents. In addition, it encompasses a very limited range of document types and layouts, being restricted primarily to single-column formats. This underscores the lack of structurally rich, diverse layouts and end-to-end benchmarks for Arabic documents and highlights the motivation for developing a more comprehensive evaluation framework.

# 3 ARABIDOC BENCHMARK

Our benchmark is designed to capture the wide spectrum of challenges introduced by complex page layouts and heterogeneous content types. Unlike previous efforts that primarily target Arabic text, our approach jointly considers multiple modalities, including charts, tables, and free-form text, within a single extraction pipeline. To construct the dataset, we first crawled document pages from diverse domains, including finance, economics, executive reports, and strategic planning documents, thereby ensuring a representative coverage of real-world use cases. From this pool, we manually reviewed and selected 137 pages that encompass a variety of content types and layouts, ensuring coverage of different combinations of textual and visual elements within a single page. Page layouts are then automatically segmented using the layout segmentation model (Zhao et al., 2024), which provides fine-grained detection of distinct objects (crops) within each page. These detected objects are first manually reviewed and refined to correct inaccurate bounding boxes and to merge those that are overly fragmented (e.g., list items, text blocks, and title/subtitle objects). This step helps simplify the representation and abstract away unnecessary granularity, thereby easing downstream reading-order verification. The refined layouts are then passed to the expert annotation model, which assigns semantic labels to every object in the original image following the predefined schema, shown in Figure 2. Finally, we aggregate the annotated crops into a structured represen-

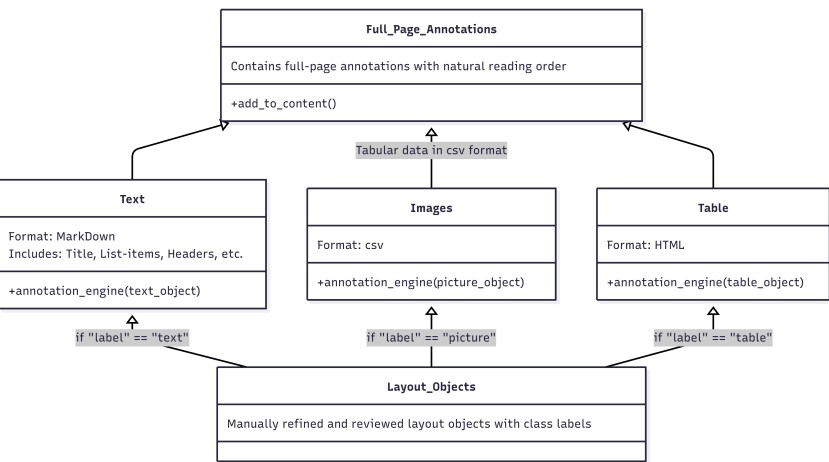

Figure 2: The diagram illustrates the annotation formatting pipeline for structured document content. Layout objects, manually refined and labeled, are routed to specialized annotation engines based on their class: text is formatted in Markdown (supporting titles, headers, lists, etc.), images (charts and diagrams) are annotated as CSV, and tables are exported in HTML.

tation that preserves both the natural reading order and the semantic content of the document. An overview of the entire data collection and annotation pipeline is presented in Figure 3.

We collected a balanced and diverse dataset of 137 document pages, consisting of 68 pages in Arabic and 69 in English, each paired with its corresponding annotations (see **Appendix A** for examples). This bilingual design not only enables direct benchmarking of Arabic document parsing but also facilitates assessing whether models, originally trained on a vast amount of English data, preserve their performance when fine-tuned on Arabic-oriented data. In this way, our benchmark can serve as a proxy for evaluating both the reliability of the model and the consistency of its original performance.

## 3.1 DATASET ANNOTATION

Here we provide a more detailed description of the annotation process. As outlined earlier, layout analysis is first applied to the document pages. Following layout analysis, we manually refine and validate each layout object, ensuring that the bounding boxes are accurate and consistent. During this refinement, we also reduce unnecessary fragmentation across multiple objects, which helps simplify the reading order and decreases complexity for downstream processing. Then each detected

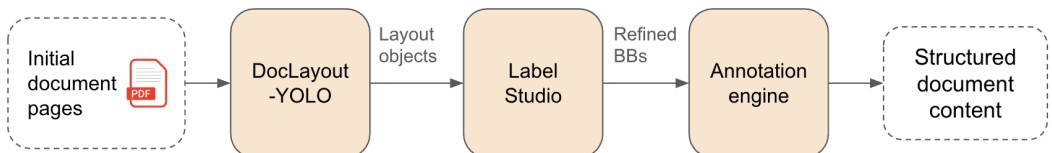

Figure 3: Overview of the dataset construction pipeline. We begin with initial document pages, which are processed by the DocLayout-YOLO (Zhao et al., 2024) model to detect layout objects. These layout outputs are then imported into Label Studio (Tkachenko et al., 2020-2025), where we manually refine inaccurate bounding boxes (BBs) and merge fragmented regions. The refined bounding boxes are subsequently passed to the annotation engine for content annotation. Finally, all annotated objects are combined into a structured document representation that preserves both visual layout and semantic content

region (crop) is individually forwarded to the expert annotation model (Gemini Team et al., 2023) rather than processing the entire page at once. This design choice is motivated by our observations in 4.1, which demonstrate that object-level annotation, combined with careful manual refinement, significantly improves the accuracy of content extraction by the expert model. This approach ensures that the ground-truth annotations in our benchmark are not only reliable but also consistently of high quality. Furthermore, following the automatic annotation and manual refinement processes, all annotations undergo a final verification to guarantee correctness.

## 3.2 DATASET STATISTICS

Our benchmark consists of 68 Arabic and 69 English document pages, containing 374 and 368 annotated regions, respectively. When grouped into three major categories: Text, Tables, and Pictures, the dataset reflects a rich multimodal balance. Textual content dominates both subsets, comprising roughly 74.8% of objects in Arabic and 83.6% of objects in English documents, consistent with the narrative-heavy nature of reports and scholarly documents. Tables, though fewer in count (11.8% Arabic, 6% English), are highly significant due to their large spatial footprint (with average normalized area: 0.168 across both languages), indicating dense information packaging. Pictures account for 13.4% of Arabic and 10.4% of English content, including charts, infographics, and diagrams. Average object density is comparable across languages ($\approx 5.4$ objects per page). These statistics confirm that the benchmark provides a balanced mixture of dense text, structured tabular data, and diverse visual elements, making it a robust benchmark for document parsing.

## 3.3 EVALUATION PIPELINE OVERVIEW

Our evaluation pipeline compares model predictions against ground-truth annotations in two main steps. We first assess reading order similarity, which focuses on the correct sequencing of high-level structural elements, and then parse the content into distinct modalities: tables, charts, and text, for a more granular evaluation. The parsing step uses regex-based extraction rules, identifying tables with HTML $< table >< /table >$ delimiters, charts with $< image >< /image >$ delimiters, and all remaining content as free-form text. Each modality is then evaluated with a specialized metric: TEDS for tables, ChartEx for charts, and a relaxed word-alignment metric for text. This approach ensures a comprehensive, end-to-end analysis of a model's performance by separately assessing its structural integrity and content-level accuracy.

### READING ORDER EVALUATION

This evaluation measures the accuracy of document reading order estimation by dividing content into semantic blocks that represent different elements such as text, tables, and images. These blocks are identified through content-based markers, producing comparable sequences for both ground truth and predicted orders. Similarity is then quantified using the Longest Common Subsequence (LCS) (Hirschberg, 1977) metric, which captures how well the predicted order preserves the structure of the original. Intuitively, LCS measures the length of the longest sequence of blocks that appear in the same order in both the ground truth and predicted sequences, without requiring the matching blocks

to be contiguous. The abstraction with semantic blocks offers a robust evaluation across varied document layouts while remaining tolerant to minor differences in content extraction or ordering.

### TEXT EVALUATION

To assess the quality of text predictions, we employ a relaxed word-level evaluation that accounts for both precision and recall. Instead of requiring exact string matches, we align the predicted sequence against the ground truth using a sequence matching algorithm. This allows us to capture overlapping word spans even when minor ordering or formatting differences are present. Precision reflects how much of the predicted content is correct, recall measures how much of the ground truth is recovered, and their harmonic mean (F1 score) provides a balanced view of performance. This design ensures that models are rewarded for partial correctness while still penalized for omissions and extraneous output, offering a more robust evaluation than strict exact matching.

### CHART EVALUATION

For chart evaluation, we adapted the ChartEx metric from Heakl et al. (2025), modifying it to better suit our document parsing scenario. In particular, we removed type and topic evaluations from the original implementation to focus solely on the tabular content of charts. Our adaptation introduces two main changes. First, we implemented logic to handle the headers section. In cases where column headers are missing in either the ground truth or the predicted chart, arbitrary headers are automatically assigned to ensure alignment during evaluation. Second, we introduced a run-time analysis for fuzzy column matching. Instead of relying solely on column names, which may differ due to variations in model predictions, we estimate the optimal column mapping by comparing all rows between the ground truth and predicted charts. This approach allows the metric to robustly align columns even when their names differ, ensuring a more accurate evaluation of chart content (see **Appendix B** for more details).

### TABLE EVALUATION

For table evaluation, we build on top of the TEDS implementation introduced by Heakl et al. (2025), to a customized version aligned with our end-to-end document parsing goals. In our formulation, the final score is calculated as a weighted sum of structure and content, assigning 20% importance to structural accuracy and 80% to content accuracy. Crucially, the content comparison is order-invariant; differences in row or column order are not penalized (see **Appendix C** for details). As long as the extracted table content is faithful and consistent with the ground truth, the system receives a higher score. This design reflects our central motivation: in practical applications such as retrieval-augmented generation (RAG) or context-enhanced text generation, the accuracy of the extracted content is far more important than perfectly replicating the original structure. By prioritizing content accuracy while relaxing structural constraints, our evaluation better reflects the real-world demands of document understanding.

## 4 EXPERIMENTS

We evaluate two groups of models: 1) Expert VLMs, which are purpose-built and trained specifically for document understanding and parsing; and 2) General-purpose VLMs, which are typically larger in scale and trained on broad, diverse datasets to support strong generalization across unseen tasks. In the general-purpose category, we include the Gemini 2.5 series (Comanici et al., 2025), GPT-4.1 (OpenAI et al., 2023), Qwen2.5-VL 72B (Bai et al., 2025), and InternVL3.5 38B (Wang et al., 2025). The expert category consists of Granite-Docling[1] and DotsOCR[2]. We omit pipeline-based approaches from our evaluation, as their operation does not align directly with the scope of our benchmark, and because the field is increasingly shifting toward end-to-end models.

Table 2 presents the results for full-page annotation on English document pages. While expert VLMs often advertise support for handling charts and diagrams, they are not intended to parse such elements in the context of full-page document understanding. Instead, these models are typically

---

[1]https://docling-project.github.io/docling/
[2]https://github.com/rednote-hilab/dots.ocr

Table 2: This table shows the evaluation results on English documents for both Expert and General-purpose Vision-Language Models (VLMs). The scores represent performance on full-page annotation and specific tasks like text, tables, charts, and document reading order. The highest score in each category is highlighted in bold.(↑) mean higher is better.

| Model Type | Model | Task-specific scores | | | | Full-page |
|---|---|---|---|---|---|---|
| | | *Charts* (↑) | *Tables* (↑) | *Text* (↑) | *Reading order* (↑) | **score** (↑) |
| **Expert VLMs** | Granite-Docling | NA | 0.64 | 0.90 | 0.62 | 0.72 |
| | Dots.OCR | NA | 0.91 | **0.94** | 0.77 | 0.87 |
| **General-purpose VLMs** | Qwen2.5-VL-72B | 0.66 | 0.79 | 0.88 | 0.78 | 0.78 |
| | InternVL3.5-38B | 0.65 | 0.77 | 0.86 | 0.70 | 0.75 |
| | GPT4-1 | 0.69 | 0.90 | 0.90 | 0.77 | 0.82 |
| | Gemini2.5-Pro | **0.92** | **0.95** | 0.91 | **0.86** | **0.91** |
| | Gemini2.5-Flash | 0.86 | 0.89 | 0.88 | 0.75 | 0.85 |

Table 3: Evaluation results on Arabic documents. Scores are provided for full-page annotation and separate tasks, including text, tables, charts, and document reading order.

| Model Type | Model | Task-specific scores | | | | Full-page |
|---|---|---|---|---|---|---|
| | | *Charts* (↑) | *Tables* (↑) | *Text* (↑) | *Reading order* (↑) | **score** (↑) |
| **Expert Vision Models** | Granite-Docling | NA | 0.0 | 0.40 | 0.09 | 0.16 |
| | Dots.OCR | NA | 0.77 | **0.82** | 0.56 | 0.72 |
| **Vision Language Models** | Qwen2.5-VL-72B | 0.70 | 0.54 | 0.51 | 0.35 | 0.53 |
| | InternVL3.5-38B | 0.53 | 0.53 | 0.09 | 0.14 | 0.32 |
| | GPT4-1 | 0.65 | 0.77 | 0.62 | 0.44 | 0.62 |
| | Gemini2.5-Pro | 0.77 | **0.88** | 0.76 | **0.62** | 0.76 |
| | Gemini2.5-Flash | **0.84** | 0.86 | 0.76 | 0.61 | **0.77** |

designed for standalone chart extraction, where a chart image is provided in isolation together with a chart-specific parsing prompt. In this controlled setting, they can accurately extract chart content; however, when applied to full-page parsing, expert models only detect chart bounding boxes and assign object classes, without performing full content extraction. This highlights their specialization for isolated chart tasks rather than integrated document-level parsing. Consequently, the CR score is not reported for expert models and is excluded from the computation of the **Full-page score**. Within the expert category, DotsOCR demonstrates the strongest overall performance, outperforming Granite-Docling and even surpassing several general-purpose models despite its relatively small size. Specifically, DotsOCR achieves the highest text parsing accuracy (0.94), along with strong results on table extraction (0.91) and reading order prediction (0.77).

Among general-purpose VLMs, Gemini 2.5-Pro attains the highest overall performance, achieving a full-page score of 0.91, supported by strong results across tables, charts, and reading order. GPT-4.1 (0.82) and Gemini 2.5-Flash (0.85) also deliver competitive performance, while Qwen2.5-VL-72B (0.78) and InternVL3.5-38B (0.75) obtain slightly lower but stable scores across evaluation dimensions.

Table 3 reports results on Arabic documents. Compared to English, both specialist and general-purpose models exhibit a substantial performance decline when processing Arabic documents, reflecting the inherent complexity of the Arabic language and its distinctive right-to-left reading order. Among expert systems, Granite-Docling performs notably poorly (full-page score 0.16), likely reflecting limited Arabic coverage in its training data; its outputs exhibit repetition and hallucinated content, which heavily suppresses accuracy. DotsOCR, although also impacted, attains a substantially higher score (0.72), indicating stronger robustness to multilingual and structurally complex pages.

Flagship proprietary general-purpose models such as GPT-4.1 (0.62) and Gemini 2.5 (0.76–0.77) exhibit consistently strong performance, indicating robust cross-lingual generalization even for low-resource languages like Arabic. While open-source models continue to show progress, their performance on Arabic documents remains mixed. InternVL3.5 performs poorly, achieving a full-page score of only 0.32. Qualitative analysis reveals frequent repetitions and the insertion of incorrect language symbols, which substantially degrade output quality. In contrast, Qwen2.5-VL demonstrates mid-range but still competitive performance, with a full-page score of 0.53, indicating a comparatively stronger ability to adapt to Arabic content despite the challenges.

### 4.1 ABLATION STUDY

**Combining Layout detection with content parsing**: We evaluated content parsing accuracy by comparing two approaches: full-page processing with region-of-interest (ROI) specification versus pre-cropped input processing. Both methods employed identical annotation models and bounding box coordinates. We assessed performance using average Word Error Rate (WER) across a sample of 10 document pages (5 Arabic, 5 English). Our findings demonstrate that pre-cropped inputs consistently deliver more accurate performance compared to full-page ROI processing. For Arabic documents, cropped inputs reduced WER from 0.563 to 0.368, representing a 35% relative improvement, where the relative improvement is calculated as:

$$\text{Relative Improvement} = \frac{1}{N} \sum_{i=1}^{N} \frac{\text{WER}_{\text{full-page}}^{(i)} - \text{WER}_{\text{crop}}^{(i)}}{\text{WER}_{\text{full-page}}^{(i)}}$$

The enhancement was even more dramatic for English documents, where WER decreased from 1.858 to 0.159, a substantial 91% relative improvement. This performance gap was particularly evident in English pages containing extensive text blocks, where cropped processing achieved a WER of 0.205 while full-page processing degraded to 3.15 due to content leakage. Analysis of the full-page ROI approach revealed three systematic failure modes that contribute to elevated error rates. First, text duplication frequently occurs, with identical content appearing multiple times in the output. Second, the model fails to respect explicitly defined ROI boundaries, leading to the inclusion of irrelevant content from outside the target region. Third, text fragments become disordered during extraction, disrupting the logical sequence of the original content. These structural errors are largely eliminated when using pre-cropped inputs, as the model operates within clearly defined spatial constraints. Based on these results, we recommend using cropped annotations whenever feasible, as this approach significantly enhances recognition accuracy while reducing structural parsing errors compared to ROI-based extraction from full pages.

## 5 CONCLUSION

Our evaluation highlights the current state of both expert and general-purpose vision-language models in full-page document parsing across English and Arabic. Expert models such as DotsOCR demonstrate competitive results, in some cases rivaling much larger general-purpose models, underscoring the value of specialized training. These promising results further highlight the usefulness and strength of task-oriented small vision models, which can achieve strong performance despite their modest scale. At the same time, proprietary general-purpose models like Gemini 2.5 and GPT-4.1 deliver consistently strong results, showing robust cross-lingual generalization. However, performance on Arabic documents remains substantially lower across all systems, with open-source models in particular, such as InternVL3.5 and Granite-Docling, struggling with repetition, symbol errors, and loss of structural coherence.

This benchmark opens a discussion on the limitations of both open-source and closed-source models when applied to full-page document parsing tasks. Even in English, challenges persist, while in Arabic the performance gap is more pronounced. These findings emphasize that parsing full-page content in an end-to-end manner is a complex task requiring more than localized text recognition. Models must accurately capture reading order, identify and differentiate structural components such as tables and charts, and preserve logical consistency throughout the page. Addressing these challenges will be central to advancing the next generation of document understanding models capable of robust multilingual and multimodal parsing.

## LIMITATIONS AND FUTURE WORK

While ArabiDoc provides the first comprehensive bilingual benchmark for full-page document parsing, it is not without limitations. First, the dataset size (137 pages) is relatively modest, which may restrict the ability of large-scale models to fully demonstrate their generalization capacity. Second, our evaluation pipeline currently emphasizes text, tables, and charts, but does not yet extend to other important modalities such as equations, forms, or handwritten annotations, which remain common in real-world documents. Third, although our bilingual design covers Arabic and English, it does not yet support other low-resource languages that would further broaden cross-lingual evaluation.

Looking ahead, we plan to expand ArabiDoc in both scale and scope. This includes increasing the dataset size with more diverse domains, incorporating additional content types such as forms and mathematical expressions, and extending coverage to other low-resource languages beyond Arabic. Moreover, we aim to refine the evaluation pipeline by integrating semantic similarity metrics and user-centric evaluations that better capture practical usability. Finally, releasing a public leaderboard and challenge track will encourage community participation and foster further progress in robust multilingual and multimodal document understanding.

## REPRODUCIBILITY STATEMENT

To ensure transparency and reproducibility, we will release all of our evaluation scripts, evaluation datasets, and corresponding annotations. The benchmark will be made publicly available, and we plan to provide an evaluation server to facilitate testing of future models or methods that were not covered within the scope of this paper.

## LLM USAGE STATEMENT

We acknowledge that we leveraged large language models (LLMs) to refine and polish the writing for improved clarity and readability. However, all research ideas, methodological designs, and implementations were fully conceived and carried out by the authors.

## ETHICS STATEMENT

This work complies with the ICLR Code of Ethics.[3]

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
