# ARABIDOC: A HOLISTIC ARABIC-ENGLISH EVALUATION SUITE FOR END-TO-END DOCUMENT PROCESSING

# SUPPLEMENTARY MATERIALS

# 1 APPENDIX A: FULL-PAGE ANNOTATIONS

(a) Original English document image

(b) Extracted structured content

Figure 1: Example of full-page annotation for an English document.

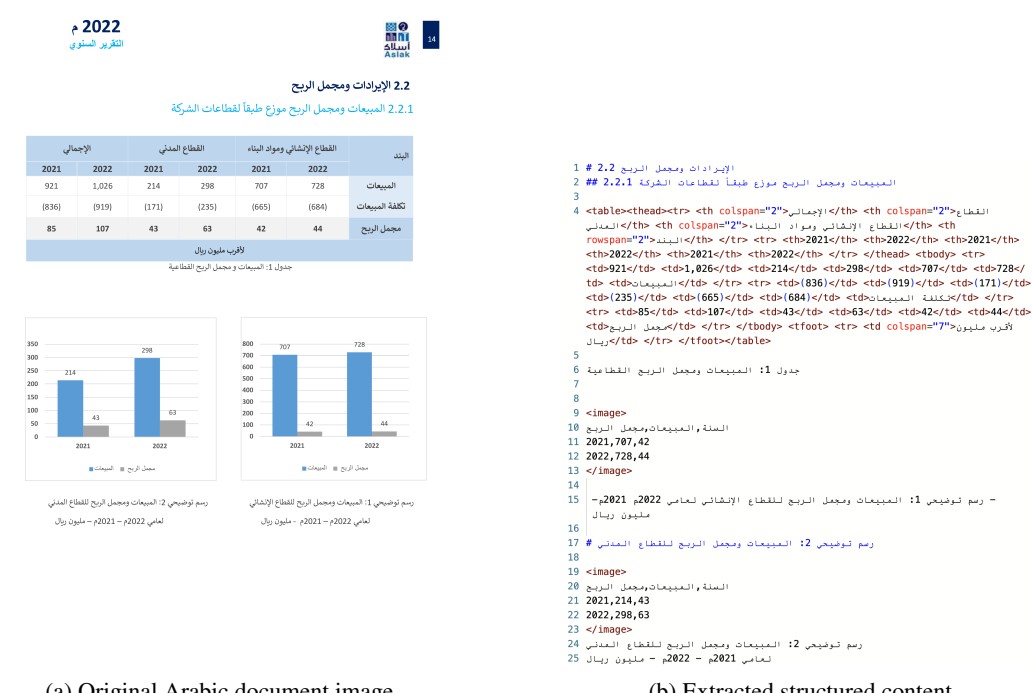

(a) Original Arabic document image      (b) Extracted structured content

Figure 2: Example of full-page annotation for an Arabic document.

# 2 APPENDIX B: CUSTOM IMPLEMENTATION OF CHARTEX METRIC

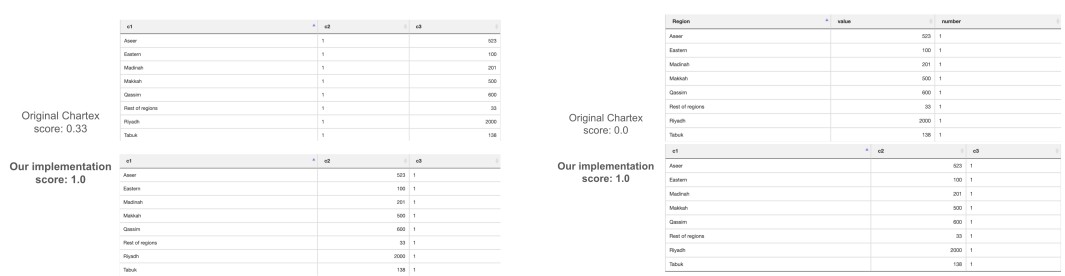

(a) Robustness to column order in CSV extraction.      (b) Column mapping via content comparison.

Figure 3: Comparison of our implementation with Chartex. (a) shows robustness to column order; despite a reordering of numerical columns c2 and c3 in our implementation, the accurate content extraction still yields a perfect score of 1.0, unlike the original Chartex's score of 0.33, which is sensitive to column permutations. (b) demonstrates improved column mapping via content comparison. The original Chartex approach receives a score of 0.0 because it relies on header names during fuzzy column matching. Our implementation, however, successfully maps the columns by comparing their content, resulting in a perfect score of 1.0, highlighting its robustness.

# 3 Appendix C: Custom implementation of Table evaluation metric

Table 1:

| Main indicators | Units | Quantity, number and value |
|---|---|---|
| Number of cows in cows specialized farms | Numbers | 411,936 |
| Quantity of raw milk produced in cows specialized farms | billion liters | 2.8 |
| Value of table eggs in layers farms | Billion SARs | 2.2 |
| value of broilers sold production | Billion SAR | 9.6 |
| Quantity of fish production in fish farming specialized farms | Ton | 738 |
| Quantity of fish production in fish farming specialized farms | Number | 6,243 |
| Quantity of fish production in fish farming specialized farms | Thousand tons | 81.3 |
| Value of sold chicks in Hatchery projects | Billion SAR | 1.6 |

Original TEDS
score: 0.875

Our implementation
score: 1.0

Table 2:

| Main indicators | Units | Quantity, number and value |
|---|---|---|
| Number of cows in cows specialized farms | Numbers | 411,936 |
| Quantity of raw milk produced in cows specialized farms | billion liters | 2.8 |
| Value of table eggs in layers farms | Billion SARs | 2.2 |
| value of broilers sold production | Billion SAR | 9.6 |
| Quantity of fish production in fish farming specialized farms | Thousand tons | 81.3 |
| Value of sold chicks in Hatchery projects | Billion SAR | 1.6 |
| Quantity of fish production in fish farming specialized farms | Ton | 738 |
| Quantity of fish production in fish farming specialized farms | Number | 6,243 |

(a) Row-agnostic evaluation.

Table 1:

| Main indicators | Quantity, number and value | Units |
|---|---|---|
| Number of cows in cows specialized farms | 411,936 | Numbers |
| Quantity of raw milk produced in cows specialized farms | 2.8 | billion liters |
| Value of table eggs in layers farms | 2.2 | Billion SARs |
| value of broilers sold production | 9.6 | Billion SAR |
| Quantity of fish production in fish farming specialized farms | 738 | Ton |
| Quantity of fish production in fish farming specialized farms | 6,243 | Number |
| Quantity of fish production in fish farming specialized farms | 81.3 | Thousand tons |
| Value of sold chicks in Hatchery projects | 1.6 | Billion SAR |

Original TEDS
score: 0.786

Our implementation
score: 1.0

Table 2:

| Main indicators | Units | Quantity, number and value |
|---|---|---|
| Number of cows in cows specialized farms | Numbers | 411,936 |
| Quantity of raw milk produced in cows specialized farms | billion liters | 2.8 |
| Value of table eggs in layers farms | Billion SARs | 2.2 |
| value of broilers sold production | Billion SAR | 9.6 |
| Quantity of fish production in fish farming specialized farms | Ton | 738 |
| Quantity of fish production in fish farming specialized farms | Number | 6,243 |
| Quantity of fish production in fish farming specialized farms | Thousand tons | 81.3 |
| Value of sold chicks in Hatchery projects | Billion SAR | 1.6 |

(b) Column-agnostic evaluation.

Figure 4: Advantages of our evaluation metric. (a) shows robustness to row reordering; while the original metric assigns a score of 0.875, our implementation correctly evaluates the content, leading to a score of 1. This highlights its robustness in ensuring accurate content assessment. (b) shows robustness to column reordering. The original metric assigns a score of 0.786, while our implementation achieves a perfect score of 1, demonstrating stronger robustness. In both cases, our implementation consistently achieves the correct score of 1.0, unlike the original metric.