# OpenReview forum: "ArabiDoc: A Holistic Arabic-English Evaluation Suite for End-to-End Document Processing"
_ICLR.cc/2026/Conference — ICLR 2026 Conference Withdrawn Submission_

### Official Review · Reviewer_T7sy · 2025-10-22

**Soundness:** 2
**Presentation:** 1
**Contribution:** 2
**Rating:** 2
**Confidence:** 5

**Summary:**

This paper introduces ArabiDoc, a bilingual (Arabic-English) benchmark for evaluating end-to-end document parsing models. The benchmark consists of 137 pages (68 Arabic, 69 English) annotated with text, tables, charts, and reading order information. The authors evaluate both expert document parsing models (Granite-Docling, DotsOCR) and general-purpose vision-language models (Gemini 2.5, GPT-4.1, Qwen2.5-VL, InternVL3.5) on this benchmark. The evaluation employs relaxed metrics including LCS for reading order, TEDS for tables, ChartEx for charts, and word-level F1 for text. Results show that models perform substantially worse on Arabic documents compared to English, with general-purpose models like Gemini 2.5-Pro achieving the best overall performance.

**Strengths:**

1. The benchmark addresses an important gap by providing end-to-end evaluation that jointly considers text, tables, and charts within full-page documents, moving beyond isolated subtask evaluation.

2. The benchmark includes reading order evaluation which is valuable for assessing holistic document understanding, particularly for Arabic's right-to-left directionality.

3. The relaxed metrics for tables (order-invariant) and text (word-level alignment) better reflect real-world requirements than strict exact matching.

**Weaknesses:**

1. **Insufficient dataset scale:** With only 137 pages, the benchmark is too small to serve as a robust evaluation standard or support meaningful statistical analysis. This severely limits its utility for assessing model generalization. Compared to KITAB-Bench (Arabic) with 8,809 samples and OmniDocBench (English) with 981 samples.

2. **Lack of diversity and fine-grained analysis:** The paper provides no breakdown of:
- Page layouts (single vs. multi-column)
- Document types (financial reports, academic papers, presentations, books)
- Table complexity (spanning cells, embedded images)
- Chart types (line, bar, pie, etc.)
- Font variations (Ruq'ah, Naskh for Arabic)
- Image quality attributes (scanned vs. digital, watermarks, tilt)

Without these statistics, it's impossible to assess the benchmark coverage or identify specific failure modes.

3. **Missing comparative analysis:** Section 3.2 claims the benchmark provides "a balanced mixture" but offers no comparison to existing benchmarks (OmniDocBench, KITAB-Bench). Comparative statistics are essential to justify design choices.

4. **Incomplete model evaluation:**

- Pipeline-based methods entirely omitted without justification
- Granite-Docling chart evaluation missing despite the model producing bounding boxes (charts could be cropped and re-fed for annotation extraction)
- No qualitative examples or failure analysis explaining why models perform poorly on Arabic

5. **Limited scope** compared to OmniDocBench: Missing formula recognition, which is critical for academic and scientific documents.

6.**Vague methodology descriptions:**

- Line 277: "sequence matching algorithm" not specified
- Lines 213-215: Refinement criteria and its impact on reading order not detailed
- Line 290-292: Unclear how fuzzy matching differs from KITAB-Bench implementation


5. **Insufficient metric justification:** Text evaluation uses word-level alignment but provides no examples demonstrating why this is superior to alternatives or how it handles Arabic morphology.

6. **No qualitative analysis:** Paper lacks examples from either the benchmark or model predictions in the main text, making it difficult to understand annotation quality or model failure patterns.

7. **Shallow analysis:** Results tables show scores but provide minimal insight into what causes performance degradation (hard fonts? scanned quality? complex layouts?). The ablation study (Section 4.1) is the only analytical component.

**Questions:**

1. **Code-switching**: Does the benchmark include code-switched Arabic-English content common in modern Arabic documents?

2. **Refinement process** (Lines 213-215): What specific criteria guide the manual refinement of bounding boxes and object merging? How does this refinement quantitatively improve reading order accuracy?

3. **Picture annotations** (Line 247): You mention pictures can include infographics. Are infographics fully annotated? What about other picture types (natural images, logos, screenshots), are they annotated or ignored?

4. **Fuzzy matching** (Lines 290-292): How does your fuzzy column matching algorithm differ from KITAB-Bench's approach? What makes it more suitable for your scenario?

5. **Granite-Docling charts**: Since the model outputs chart bounding boxes, why not crop and re-feed these regions to evaluate chart parsing capability?

6. **Evaluation completeness**: Why were pipeline-based methods excluded? They remain widely used and could provide important baselines.

---

### Official Review · Reviewer_xj4s · 2025-10-30

**Soundness:** 2
**Presentation:** 1
**Contribution:** 1
**Rating:** 2
**Confidence:** 4

**Summary:**

This paper proposes an arabic-english document parsing evaluation benchmark.

**Strengths:**

This work introduces a new benchmark for Arabic document analysis and makes a certain contribution. However, its overall impact is restricted by limitations in areas like dataset quality, which constrains the scope of its contribution.

**Weaknesses:**

1. The main text does not meet the requirement of 9 pages.

2. Too few evaluation methods are included. Too many mainstream multilingual OCR solutions (e.g., mineru, olmocr, nanonets, etc.) have not been evaluated.

3. No results for formulas are provided in the task-specific section.

4. Results are not categorized by different document types.

5. The dataset analysis and visualization results are insufficient, making it difficult to demonstrate the contribution of the dataset. The overall contribution of the work is not convincing. How can we verify the diversity of Arabic documents in the evaluation set? How can we understand the performance differences among different models?

**Questions:**

see weakness

---

### Official Review · Reviewer_S7nn · 2025-10-31

**Soundness:** 1
**Presentation:** 2
**Contribution:** 2
**Rating:** 2
**Confidence:** 5

**Summary:**

Existing low-resource language (e.g., Arabic) document benchmarks lack comprehensive full-document parsing evaluation, so this work proposes an Arabic-English bilingual benchmark integrating diverse elements for end-to-end parsing. With 137 human-annotated pages and three key features (reading order preservation, multi-type visual content support, relaxed metrics), it establishes the first comprehensive standard for structured Arabic document parsing and cross-lingual evaluation.

**Strengths:**

The paper shows originality in addressing low-resource (e.g., Arabic) document parsing gaps with a bilingual framework, and high quality through rigorous human-annotated data.

**Weaknesses:**

I strongly recommend that the authors rewrite this paper.
1. What I truly fail to comprehend is that over a dozen OCR VLMs have emerged in the past six months, yet the authors only evaluated six models (including general-purpose VLMs)—which is simply inadequate for a benchmark.
2. Additionally, why have the authors failed to provide even a single sample image of the dataset?
3. Furthermore, is there any real necessity to incorporate English documents into the dataset?

**Questions:**

see the weakness

---

### Official Review · Reviewer_T5vW · 2025-10-31

**Soundness:** 1
**Presentation:** 4
**Contribution:** 2
**Rating:** 2
**Confidence:** 2

**Summary:**

This paper introduces ArabiDoc, a new bilingual (Arabic-English) benchmark dataset for end-to-end document processing. The authors' goal is to address a gap in existing benchmarks, which they argue are often fragmented into single-task evaluations (like OCR or table recognition) and lack support for low-resource languages like Arabic.

The benchmark consists of 137 document pages (68 Arabic, 69 English) sourced from finance and executive reports. Its key contributions are:

1. Holistic Evaluation: It assesses the parsing of diverse elements—text, tables, and charts—within a single, unified framework.

2. Reading Order: It explicitly preserves and evaluates the natural reading order of document elements.

3. Relaxed Metrics: It proposes evaluation metrics that prioritize "practical usability" over "strict exactness". For example, the table metric (a modified TEDS) heavily weights content accuracy (80%) over structural accuracy (20%) , reflecting the needs of downstream tasks like Retrieval-Augmented Generation (RAG).


The paper evaluates several "expert" and "general-purpose" VLMs, concluding that while modern models like Gemini 2.5 show strong performance, a significant performance gap persists, especially on Arabic documents.

**Strengths:**

1.  **Problem Identification:** The paper correctly identifies a clear and important gap in the literature: the lack of comprehensive, end-to-end document parsing benchmarks, especially for low-resource languages like Arabic.
2.  **Pragmatic Metric Philosophy:** The authors' argument for "relaxed" metrics that prioritize content accuracy for downstream tasks (like RAG) is a strong, pragmatic, and welcome contribution. The modified TEDS score (80% content, 20% structure) is a good example of this.
3.  **Cross-Lingual Analysis:** The bilingual nature of the dataset (Arabic and English) allows for a direct and valuable comparison of model performance, clearly highlighting the cross-lingual generalization gap.
4.  **Ablation Study Finding:** Though it contradicts the paper's main premise, the finding in Section 4.1 that pre-cropped inputs are "dramatically" (91% relative improvement in English) better than full-page ROI parsing is a significant and actionable insight.
5.  **Clear Presentation:** The paper is well-written, clear, and easy to follow. The motivation for the work is well-argued.
6.  **Helpful Visuals:** Figures like the radar chart (Figure 1), the annotation pipeline (Figure 2), and the dataset construction diagram (Figure 3) are helpful for understanding the evaluation and data creation process. Table 1 also provides a useful comparison against prior work.

**Weaknesses:**

1.  **Critically Insufficient Dataset Size:** The primary weakness is that **137 pages is not a benchmark**; it is a small test set. This size is insufficient to capture the diversity of "complex real-world documents" or to provide a stable, generalizable measure of VLM performance. The authors themselves acknowledge this as a limitation.
2.  **Fatal Annotation Bias:** The decision to use an "expert annotation model", specifically Gemini, to generate the ground-truth annotations is a fundamental methodological flaw. This creates an unmitigated conflict of interest when Gemini models (Gemini 2.5-Pro/Flash) are then evaluated and achieve the highest scores. The paper provides no quantitative analysis of the "manual verification" step to prove that this bias was removed.
3.  **Contradictory Narrative and Flawed Main Experiment:** The paper promotes "holistic" and "end-to-end" parsing, yet its own ablation study (Section 4.1) proves this approach is deeply flawed and **vastly inferior to a modular detect-and-crop pipeline**. The paper fails to reckon with this contradiction and proceeds to use the inferior full-page method for its main experiments, rendering the results in Tables 2 & 3 uninformative of *actual* SOTA capability.
4.  **Superficial Evaluation of "Reading Order":** The evaluation of reading order via the Longest Common Subsequence (LCS) is overly simplistic. It treats the document as a linear sequence, ignoring the far more complex *logical* and *hierarchical* structure (e.g., 'this is a caption for that table,' 'this list belongs to this header'). This misses a key component of true document understanding.
5.  **Weak "Relaxed" Text Metric:** For a paper that (correctly) champions pragmatic metrics for RAG, the choice of a "relaxed word-alignment metric" is weak. For RAG, *semantic* equivalence is what matters, not lexical overlap. A metric like BERTScore would have been far better aligned with the paper's stated philosophy.
6.  **Buried Key Finding:** The paper's most significant finding—the superiority of the detect-and-crop pipeline—is buried in an "Ablation Study" (Section 4.1) rather than being a central part of the results and discussion.

**Questions:**

1.  **On Annotation Bias:** Can you provide a quantitative analysis of the "manual verification" step? What percentage of the "expert annotation model's" outputs were corrected by human annotators? What were the common error types? How can you assure the community that the final ground truth is free of "Gemini-flavor" artifacts?
2.  **On Dataset Creation:** Why was a competitor model (e.g., GPT-4.1, Claude 3) not used to generate a portion of the annotations, or why was a fully-manual-from-scratch annotation process not used on a subset, to measure and control for this obvious annotator-model bias?
3.  **On Experimental Setup:** Your ablation study (Section 4.1) provides strong evidence that a detect-and-crop pipeline is "dramatically" better than the full-page parsing method used in your main experiments (Tables 2 & 3). Why did you not use this superior pipeline-based approach for your main evaluation? The current results seem to evaluate a sub-optimal "strawman" version of these models.
4.  **On Dataset Scale:** How can the authors justify that 137 pages are sufficient to form a "benchmark" that can produce reliable and generalizable conclusions, especially when the authors themselves state in their limitations that this modest size "may restrict the ability of large-scale models to fully demonstrate their generalization capacity"?
5.  **On Metric Choice:** Given the stated focus on "practical usability" for RAG, why did you choose a "word-alignment" metric for text evaluation instead of a *semantic* similarity metric (e.g., BERTScore), which would better capture whether the *meaning* was extracted correctly?

**Details Of Ethics Concerns:**

The paper uses an "expert annotation model", which it identifies as a Gemini model, to generate the ground-truth data for the benchmark. It then evaluates other Gemini models (Gemini 2.5 Pro/Flash) on this same benchmark, where they achieve the highest scores.

This represents a significant potential conflict of interest and a "data contamination" or benchmark bias issue. The benchmark may be measuring similarity to the annotation model rather than true task performance. This is a responsible research practice concern, as the validity and fairness of the benchmark are questionable. While the authors disclose using LLMs for *writing*, the use of an LLM for *ground-truth generation* is a far more significant methodological issue that is not adequately transparently addressed or mitigated.

---

### Note · Authors · 2025-11-16

I have read and agree with the venue's withdrawal policy on behalf of myself and my co-authors.